# The Effect of Sc and Zr Additions on the Structure, Mechanical, and Corrosion Properties of a High Thermal Conductive Al–3%Zn–3%Ca Alloy

**DOI:** 10.3390/ma18245680

**Published:** 2025-12-18

**Authors:** Anastasia Lyskovich, Viacheslav Bazhenov, Ivan Baranov, Mikhail Gorshenkov, Olga Voropaeva, Andrey Stepashkin, Vitaliy Doroshenko, Ruslan Yu. Barkov, Shevket Rustemov, Andrey Koltygin

**Affiliations:** 1Casting Department, National University of Science and Technology “MISIS”, Leninskiy pr. 4, 119049 Moscow, Russia; lyskovich.aa@misis.ru (A.L.); baranov.ii@misis.ru (I.B.); rustemov.sh.kh@gmail.com (S.R.); misistlp@mail.ru (A.K.); 2Physical Materials Science Department, National University of Science and Technology “MISIS”, Leninskiy pr. 4, 119049 Moscow, Russia; mvg@misis.ru; 3Department of Metallurgy Steel, New Production Technologies and Protection of Metals, National University of Science and Technology “MISIS”, Leninskiy pr. 4, 119049 Moscow, Russia; oovoropaeva@mail.ru; 4Center of Composite Materials, National University of Science and Technology “MISIS”, Leninskiy pr. 4, 119049 Moscow, Russia; a.stepashkin@yandex.ru; 5Department of Metal Forming, National University of Science and Technology “MISIS”, Leninskiy pr. 4, 119049 Moscow, Russia; v.doroshenko@mail.ru; 6Department of Physical Metallurgy of Non-Ferrous Metals, National University of Science and Technology “MISIS”, Leninskiy pr. 4, 119049 Moscow, Russia; barkov@misis.ru

**Keywords:** aluminum alloys, thermal conductivity, phase composition, Sc addition, Zr addition, heat treatment, tensile test, corrosion resistance

## Abstract

Al–Zn–Ca alloys are good candidates for industrial electronics and electric vehicles due to their high thermal conductivity, castability, and corrosion resistance, but their strength requires improvement. This study investigates how Sc and Zr additions affect the microstructure, thermal, mechanical, and corrosion properties of an Al–3 wt% Zn–3 wt% Ca base alloy. Microstructural analysis showed that substituting Sc with Zr did not drastically alter the phase composition but changed the elemental distribution: Sc was uniform, while Zr segregated to center of dendritic cell. Zr addition also refined the grain size from 488 to 338 μm. An optimal aging treatment at 300 °C for 3 h was established, which enhanced hardness for all alloys via precipitation of Al_3_Sc/Al_3_(Sc,Zr) particles. However, this Zr substitution reduced thermal conductivity (from 184.7 to 168.0 W/mK) and ultimate tensile strength (from 269 to 206 MPa), though it improved elongation at fracture (from 4.6 to 7.1%). All aged alloys exhibited high corrosion resistance in 5.7% NaCl + 0.3% H_2_O_2_ water solution, with Zr-containing variants showing a lower corrosion rate and better pitting resistance. The study confirms the potential of tuning Sc/Zr ratios in Al–Zn–Ca alloys to achieve a favorable balance of strength, ductility, thermal conductivity, and corrosion resistance.

## 1. Introduction

In the context of a growing demand for electrical appliances and electric vehicles, the necessity for effective heat sinks is increasing. These devices are tasked with maintaining optimal heat dissipation during operation to prevent overheating. The materials used in the construction of contemporary cooling heat sinks must exhibit high thermal conductivity in order to facilitate rapid and uniform heat transfer, thereby ensuring the optimal operation of electrical equipment [1]. In the development of a highly thermally conductive aluminum alloy, great care is taken to ensure an optimal combination of thermal conductivity and mechanical properties.

In order to meet the requirements of modern heat sinks, it is necessary for them to combine low weight with high heat dissipation capacity. The following design solutions are employed in the creation of heat sinks: (i) the number of fins is typically augmented to enhance thermal regulation efficiency; (ii) the product’s weight is reduced by reducing the thickness of the fins and the base of the heat sink [2]. In light of the intricate design of heat sinks, the predominant manufacturing technique employed for their large-scale production is die-casting [3,4].

Commercial casting aluminum alloys typically exhibit either high castability and strength or good thermal conductivity. This is attributed to the fact that aluminum alloys with good castability contain large amounts of alloying elements such as Si, Mg, and Cu. When present in high concentrations, these elements significantly impair the alloys’ thermal conductivity [5,6]. A large number of studies have been dedicated to addressing this issue [7,8,9,10,11]. It has been demonstrated that heat treatment of alloys can improve their thermal conductivity [12,13,14]. However, this outcome is frequently inadequate in ensuring effective heat transfer, owing to the initially low thermal conductivity of commercial cast alloys [15]. Alternative approaches involve incorporating reinforcements with high intrinsic thermal conductivity, such as graphene nanoplatelets (~5000 W/mK), into aluminum matrix [16,17,18]. This method can significantly improve the electrical and thermal conductivity of aluminum alloys. For example, Andilab et al. [17] succeeded in increasing the electrical conductivity of cast A319 Al alloy from 27 to 30.2% IACS after adding 0.2 wt% graphene. However, this technique is not yet fully explored, and potential adverse effects on castability, essential for manufacturing thin-walled die-cast components, remain a concern. Therefore, a key strategy for developing aluminum alloys with high thermal conductivity involves the selection of alloying elements with extremely low solubility in the α-Al solid solution or controlling the overall solubility of alloying elements in the alloy [19].

Among the promising candidates, the Al–Ca alloys are of particular interest, containing the α-Al + Al_4_Ca eutectic with a very fine structure [20,21]. The Al–Ca base alloys’ advantage is their favorable castability [22], and the virtually negligible solubility of Ca in α-Al [23]. In the preceding study, the Al–3 wt% Zn–3 wt% Ca alloy exhibited considerable potential for utilization as a heat-conducting material for the fabrication of heat sinks through the casting technique [24]. Nevertheless, despite the alloy’s high thermal conductivity (approximately 194 W/mK) and favorable castability and corrosion resistance, its tensile yield strength remained below 50 MPa. The low strength of the alloy represents a significant limitation on its potential applications.

One of the most effective methods for enhancing the mechanical properties of aluminum alloys is the alloying with rare earth elements, with Sc being the most efficacious [25,26]. During the solidification of Al–Sc alloys, Al_3_Sc particles are precipitated, which have a face-centered cubic lattice of the L1_2_ type [27]. These particles serve as nucleation sites in the aluminum melt during solidification, thereby contributing to grain refinement [28,29,30]. The precipitates of the Al_3_Sc phase formed during heat treatment serve as a strong hardener of aluminum alloys [27]. Nevertheless, the high cost of Sc makes its use in alloys for large-scale production of castings challenging. Consequently, in order to reduce the cost of the material, Sc is frequently employed in conjunction with Zr. During the solidification of Al–Sc–Zr alloys, not only primary Al_3_Sc crystals but also Al_3_(Sc,Zr) crystals, which also have an L1_2_ structure, precipitated [31,32]. Furthermore, nanosized Al_3_(Sc,Zr) precipitates markedly impede the movement of dislocations and grain boundaries, thereby enhancing the strength of the alloy [33]. Accordingly, the addition of Sc and Zr may facilitate the attainment of superior mechanical properties in the promising Al–3 wt% Zn–3 wt% Ca aluminum alloy.

A substantial body of research has been dedicated to investigating the impact of Sc and Zr on the structure and mechanical properties of aluminum alloys. However, a comprehensive analysis of the influence of these elements on Al–Zn–Ca alloys, as well as their effect on the alloys’ thermal conductivity, remains absent from the extant published literature, indicating a significant gap and novelty. As a result, the objective of the study is to ascertain the effect of the Sc and Zr ratio in the Al–3 wt% Zn–3 wt% Ca alloy on the mechanical properties, corrosion properties, and thermal conductivity and to determine the optimal heat treatment regime for mentioned alloys.

## 2. Materials and Methods

Four Al–3 wt% Zn–3 wt% Ca–Sc–Zr alloys with different Sc/Zr ratios were prepared. The melting was carried out in a clay–graphite crucible using a high-frequency induction furnace. The following master alloys and pure metals were used: Al (99.99 wt%), Zn (99.98 wt%), Al–11 wt% Ca, Al–2 wt% Sc, and Al–3 wt% Zr. After the alloy was completely melted, the C_2_Cl_6_ was employed to degassing at melt temperature of 740 °C. Next, the melt was cooled down to reach the temperature of 720 °C and was poured into the permanent steel and graphite molds. The specimens for structural analysis and tensile testing were cut from the 32 × 340 × 50 mm^3^ ingots produced in the graphite mold. The specimens for hardness measurements, thermal conductivity evaluation, and corrosion resistance were cut from the ⌀35 × 150 mm ingots obtained in the steel mold.

The equilibrium (Lever rule) and non-equilibrium (Scheil–Gulliver solidification model [34]) solidification process of Al–Zn–Ca–Sc–Zr alloys and precipitation kinetics of Al_3_Sc/Al_3_(Sc,Zr) were calculated using the Thermo-Calc (version 2016a, Thermo-Calc Software AB, Solna, Sweden) with the TTAL5 and MOBAL1 databases, which provide thermodynamic and kinetic data, respectively [35].

The microstructures of the alloys and the alloys’ composition were investigated using energy-dispersive X-ray spectroscopy (EDS) on a Tescan Vega SBH3 scanning electron microscope (SEM, Brno, Czech Republic) with an Oxford EDS detector (Oxford Instruments, Oxford, UK). The as-cast alloy compositions are summarized in Table 1. Four specimens (20 × 30 × 10 mm^3^), one for each alloy, were ground and polished for metallographic observations. The polished specimens were anodized in Barker’s solution (10% HBF_4_ water solution) according to the following regime: current density 0.1 A/cm^2^; voltage: 25 V; exposure time 30–60 s. The average grain size was measured using the linear-intercept method. To analyze the average grain size, at least 3 images of the macrostructure were taken in different parts of the sample.

The samples for transmission electron microscopy (TEM) were subjected to mechanical grinding, resulting in a thickness of approximately 300 µm. Next, the foils for TEM were prepared by double-jet electropolishing in HNO_3_–CH_3_OH (1:4 volume fractions) electrolyte held at −20 °C using TenuPol 5 polishing machine (STRUERS, Ballerup, Denmark). TEM observations were performed using bright-field, dark-field imaging, and select-area electron diffraction with the help of JEM 1400 (JEOL, Tokyo, Japan) transmission electron microscope operating at 120 kV. The sizes of the precipitates are determined using ImageJ software (version 1.52a, National Institutes of Health, Bethesda, MD, USA) on the dark-field TEM images. Each average value of the precipitate diameter is deduced from a measurement of more than 400 precipitates.

The as-cast alloys were aged at 200–400 °C from 3 to 24 h. The effect of aging treatment on alloys was evaluated by analysis of hardness and thermal conductivity. The specimens, measuring 20 × 10 × 10 mm^3^ were then ground using P400 grit SiC abrasive paper. Brinell hardness was measured using an Nemesis 9001 (INNOVATEST, Maastricht, The Netherlands) universal hardness tester with the following parameters: a ball with a diameter of 2.5 mm, an applied load of 62.5 kgf (≈613 N), dwelling time of 10 s. Average hardness results were obtained from 5 repeated measurements.

To estimate the thermal conductivity, the Smith and Palmer equation was used, which establishes the correlation between thermal conductivity (λ) and electrical conductivity (*σ*) of alloys, specifically, aluminum alloys (without silicon) [36]. Poirier and McBride [37] confirmed the high degree of agreement between the results of the calculation method for determining thermal conductivity using the Smith and Palmer equation and the measurement results. The following equation was used in this study [24]:*λ* = 0.816*L*_0_*Tσ* + 17.94,(1)
where *L*_0_ is the Lorenz constant, *T* is the temperature. The Lorentz constant in Equation (1) is 2.45 × 10^−8^ WΩK^−2^ [37]. Using a contact-free eddy current conductivity meter VE-27NC (Sigma, Ekaterinburg, Russia) with a measurement range of 5.0–37.0 MS/m, the *σ* was measured at room temperature. Generally, 10 measurements were performed to evaluate the average *σ* value for each specimen. Next, the thermal conductivity was calculated using Equation (1).

To determine the temperature dependence of thermal conductivity (λ), λ was calculated from the thermal diffusivity equation:*λ* = *aρC_p_*(2)
where *ρ* is the density, *a* is the thermal diffusivity, *C_p_* is the heat capacity. The variation in density with temperature *ρ(T)* was derived through the thermal expansion coefficient, which was ascertained using a DIL 402 C dilatometer (NETZSCH, Selb, Germany). The thermal diffusivity *a(T)* was determined using the laser flash method (LFA), with the LFA 447 apparatus (NETZSCH, Selb, Germany). The variation in heat capacity of aluminum with temperature *C_p_(T)* was calculated using the NIST JANAF equations [38]. The relationships between these thermal properties and temperature were approximated using third-order polynomials.

The mechanical properties of alloys after aging at 300 °C for 3 h were obtained. For tensile testing, an 5569 universal testing machine (Instron, Norwood, MA, USA) with an advanced video extensometer was employed. Cylindrical specimens with a gauge diameter of 5 mm and a length of 80 mm were obtained by lathing [19]. The tests were conducted at a constant strain rate of 0.002 s^–1^. The mean value of at least three measurements was calculated for each specimen.

Immersion corrosion testing was conducted on as-aged specimens (300 °C, 3 h) measuring 15 × 15 × 10 mm^3^ (surface area of approximately 10.5 cm^2^) in an aqueous solution comprising 5.7% NaCl and 0.3% H_2_O_2_ at 25 °C for a period of 15 days [39]. Four specimens were tested for each alloy. Prior to the commencement of the test, the specimens were ground using P600 grit SiC abrasive paper and weighed on the balance of the HR-202i model (A&D, Tokyo, Japan). To compensate for the decomposition of H_2_O_2_ in the aqueous medium, the 0.3% H_2_O_2_ was added every five days. Upon completion of the test, the corrosion products were removed by dipping the specimens into concentrated HNO_3_ [40]. The specimens were then weighed again to determine the mass gain per unit surface area. Finally, the mean corrosion rate in mm/year was calculated in accordance with the ASTM standard [40].

The electrochemical test was performed in a 5.7% NaCl + 0.3% H_2_O_2_ water solution at 25 °C using an IPC Pro MF (Volta, St. Petersburg, Russia) potentiostat/galvanostat/FRA system configured in a three-electrode arrangement. The working electrodes consisted of alloy specimens with an exposed surface area of approximately 0.9 cm^2^. The testing surface was mechanically ground using P2000 grit SiC abrasive paper and polished. Afterwards, all specimens were cleaned in distilled water. A platinum counter electrode and a saturated Ag/AgCl reference electrode completed the electrochemical cell. Prior to the test, the working electrode was placed in the electrolyte for 50 min to obtain a stable OCP. Potentiodynamic polarization curves were recorded by scanning from −1.6 V (cathodic region) to −0.6 V (anodic region) at a scan rate of 1 mV/s. For the electrochemical tests, each specimen was tested on three occasions to ensure the reproducibility of the data and the presentation of the average curves.

## 3. Results and Discussion

### 3.1. Effect of Sc and Zr Additions on as-Cast Structure of Al–Zn–Ca Alloys

The solidification pathways of Al–Zn–Ca–Sc–Zr alloys that were calculated using Lever rule and Scheil–Gulliver solidification models are presented in Figure 1. The Scheil–Gulliver solidification calculations of the AlZn3Ca3Sc0.3 alloy indicate that, following the formation of primary α-Al crystals, an eutectic reaction, L → α-Al + L1_2_, where L1_2_ is Al_3_Sc phases having a cubic L1_2_-ordered crystal structure, should occur. The solidification of this alloy should be completed by the ternary eutectic reaction L → α-Al + L1_2_ + Al_4_Ca. As demonstrated in Figure 1b–d, the solidification process in alloys containing Zr commences with the formation of primary D0_23_ crystals, where D0_23_ is Al_3_(Sc,Zr) phase which has a tetragonal D0_23_ crystal structure. Subsequently, in alloys containing 0.2 wt% Sc and (0.05–0.1) wt% Zr at a temperature of approximately 624 °C, the L1_2_-Al_3_(Sc,Zr) phase is precipitated from the melt in accordance with the eutectic reaction. The solidification process of these alloys culminates in the formation of a eutectic structure of D0_23_ + α-Al + L1_2_ + Al_4_Ca. In the case of the as-cast alloy containing 0.1 wt% Sc and 0.2 wt% Zr, the resulting structure will consist of D0_23_ + α-Al + Al_4_Ca. The fraction of L1_2_-Al_3_Sc/Al_3_(Sc,Zr) and D0_23_-Al_3_(Sc,Zr) precipitates was calculated and shown in Figure 1e. It was found that as Sc is replaced by Zr, the amount of L1_2_ phase decreases from 0.251 to 0 at%, while the amount of D0_23_ phase increases from 0 to 0.205 at%.

Figure 2 shows the as-cast microstructure of the alloys and the EDS maps showing the alloying elements distribution in the specified region. The structure of the as-cast alloys in partial agreement with calculations in the Thermo-Calc program and consists of α-Al dendrites and α-Al + (Al,Zn)_4_Ca eutectic. Previous studies have indicated that approximately half of Zn contained in the alloys is in the Al_4_Ca phase, transforming it into (Al,Zn)_4_Ca that cannot be predicted by CALPHAD calculations [24,41]. The calculation of non-equilibrium solidification (Figure 1) indicates the presence of particles of the L1_2_ and D0_23_ phases in the cast structure of the alloys. However, these phases were not detected in SEM images due to the low amount of these precipitates.

The low Sc content likely explains the limited precipitation of L1_2_ and D0_23_ phases, which remain undetectable by EDS analysis. Lohar et al. [42] found that the precipitation of the Al_3_Sc and Al_3_(Sc,Zr) phases in the structure of as-cast alloys was confirmed only in alloys with a Sc content of 0.5 wt%. Additionally, Shi et al. [43] revealed that the diffraction peaks of the Al_3_Sc phase in as-cast alloy were undetected when the total content of Sc and Zr was approximately 0.3 wt%. There is no significant difference in the α-Al + (Al,Zn)_4_Ca eutectic structure between the Sc-containing alloy and the Sc,Zr-containing alloys. The results of the quantitative analysis of the composition of individual particles and structural components of the AlZn3Ca3Sc0.2Zr0.1 alloy are presented in Table 2. The Sc is distributed relatively uniformly in α-Al and in the eutectic region. Conversely, the amount of Zr in the eutectic region is observed to approach zero. This distribution pattern remains consistent across all Zr-containing alloys under investigation.

Figure 3 shows the macrostructure of the as-cast alloys and indicates their average grain sizes. The macrostructure of the alloys consists primarily of relatively large dendritic equiaxed grains. Maximum grain refinement was achieved in the alloy with a Sc/Zr ratio of 0.2/0.05, demonstrating an average grain size of approximately 338 μm, compared to approximately 488 μm for the AlZn3Ca3Sc0.3 alloy. The grain structure of the AlZn3Ca3Sc0.2Zr0.1 and AlZn3Ca3Sc0.1Zr0.2 alloys are virtually similar, with sizes 427 and 412 μm, respectively.

The partial replacement of Sc with Zr in all instances results in grain refinement. This suggests that the refining effect resulting from the combined use of Sc and Zr is more pronounced than the effect observed with Sc alone, despite the approximate equivalence in total amount of additives. According to the solidification pathways, the Al_3_Sc phase in the AlZn3Ca3Sc0.3 alloy precipitates only via the eutectic reaction (Figure 1a). In contrast, in alloys with Zr, solidification begins with the precipitation of primary Al_3_(Sc,Zr) particles (Figure 1b–d). Consequently, the observed grain refinement in Zr-containing alloys can be attributed to the presence of primary Al_3_(Sc,Zr) particles that act as effective heterogeneous nucleation sites for aluminum dendrites. Liu et al. [44] noted that heterogeneous nucleation is enhanced in alloys containing Zr and Sc due to the lower interface energy of α-Al/Al_3_(Sc,Zr). Similar grain refinement has been documented by other researchers [45,46].

### 3.2. Effect of Aging Treatment on Hardness and Thermal Conductivity of Alloys

The effect of aging temperature and duration on the hardness of Al–Zn–Ca–Sc–Zr alloys is shown in Figure 4. As can be seen, the hardness of the alloys in the as-cast state is almost equal and amounts to 49–52 HB. It is obvious that as the aging temperature increased from 200 to 300 °C, the hardness of the alloys increased, but a further increase in aging temperature resulted in a decrease in hardness. It is worth noting that the effect of aging temperature on hardness is similar for all alloys. When the aging temperature is 200 °C, the hardness of the alloys increases only slightly during aging and does not exceed 59 HB after 24 h of aging. During aging at 250 °C, the hardness of the alloys increases more intensively and reaches 76–90 HB after 24 h. It has been established that the primary reason for alloys strengthening during aging is the formation of Al_3_Sc/Al_3_(Sc,Zr) precipitates [47,48]. At temperatures of 300–400 °C, the maximum hardness is attained after 3 h of aging and then remains essentially unaltered for all alloys. With regard to the alloy with 0.3 wt% Sc, it is apparent that at aging temperatures of 350 and 400 °C, subsequent to attaining the peak hardness value, a decrease in hardness is observed. As demonstrated by Watanabe et al., this phenomenon can be attributed to the coarsening and growth of the Al_3_Sc particles over the aging time [49]. Concurrently, the Al_3_(Sc,Zr) phase is renowned for its thermal stability, which effectively prevents the softening of alloys during prolonged holding periods at high temperatures [50]. The following peak hardness was obtained after aging for 3 h at 300 °C: 93 HB for AlZn3Ca3Sc0.3–, 87 HB for AlZn3Ca3Sc0.2Zr0.5–, 83 HB for AlZn3Ca3Sc0.2Zr0.1–, and 72 HB for AlZn3Ca3Sc0.1Zr0.2–. It can be observed that, in general, the lower the fraction of Sc in an alloy, the lower the degree of hardening. This tendency is consistent with the results of Lohar et al., which also determined that the hardness increased with increasing Sc content in Al–Sc–Zr alloys after being annealed at 300–470 °C [42].

The change in thermal conductivity during the aging treatment at temperatures of 200–400 °C is shown in Figure 5. It was observed that an increase in the fraction of Zr in the as-cast alloys led to a decrease in thermal conductivity from 167.6 to 158.9 W/mK. In metals and alloys, thermal conductivity is primarily determined by the energy transport by free electrons. Thus, any distortion of the lattice, which can be caused by impurities, defects, or dissolved atoms in a solid solution, etc., results in a decrease in thermal conductivity [6,51]. In the present study, the predominant factor influencing thermal conductivity of alloys in the as-cast condition appears to be lattice strain caused by the saturation of the α-Al solid solution with Zr and/or Sc atoms. As exemplified by the AlZn3Ca3Sc0.2Zr0.1 alloy (Table 2), the distribution of Sc and Zr in the as-cast structure differs, with Zr content in α-Al being approximately twice that of Sc. Consequently, the observed decrease in thermal conductivity is likely due to the increased Zr content in α-Al.

As illustrated in Figure 5, the thermal conductivity of the alloys exhibited an increase with an elevation in the aging temperature. For example, when the aging time was maintained at 3 h and the aging temperature increased from 200 to 400 °C, the thermal conductivity of the AlZn3Ca3Sc0.3 alloy exhibited a notable enhancement, rising from 165.0 to 189.5 W/mK. A comparable increase in the Zr-containing alloys was observed. The observed increase in thermal conductivity during aging is likely attributable to the continuous precipitation of Al_3_Sc/Al_3_(Sc,Zr) phase, which effectively reduces the content of Sc and Zr in α-Al. It has been established that the presence of secondary phases impedes electron mobility to a lesser extent [6,51]. Moreover, the coherence of Al_3_Sc/Al_3_(Sc,Zr) particles with the matrix further mitigates their impact on electrical resistivity [52]. In consideration of the established correlation between electrical and thermal conductivity of metals as delineated by the Wiedemann–Franz law, it can be deduced that coherent Al_3_Sc/Al_3_(Sc,Zr) nanoparticles should not substantially decrease the thermal conductivity of alloys. With regard to the aging time, the outcome was analogous to that observed for hardness: the majority of alloys reached their peak thermal conductivity within a period of 3–6 h. Following this initial peak, subsequent aging had a negligible effect on hardness. In summary, it was evident that the mechanical properties and thermal conductivity of Al–Zn–Ca–Sc–Zr alloys could be optimized through the change in aging parameters. The most promising heat treatment regime was aging during 3 h at 300 °C, as this regime was found to simultaneously achieve the highest hardness and moderate thermal conductivity. This heat treatment achieved dual objectives: it restored thermal conductivity to levels approaching the base Al–3 wt% Zn–3 wt% Ca alloy (194 W/mK [24]), while increasing hardness by at least 31 HB relative to the base alloy (41 HB [24]). Furthermore, the thermal conductivity of the as-aged AlZn3Ca3Sc0.3 alloy (184.7 W/mK) is only slightly inferior to that of the Al–3.6Si–0.4Mg–0.03Ti alloy (T5–225 °C, 2 h) used by Tesla (~190.7 W/mK using Equation (1)) [53]. The thermal conductivity of pure aluminum at room temperature is 237 W/mK [54], and the thermal conductivity of the as-aged AlZn3Ca3Sc0.3 alloy is approximately 78% of the pure aluminum thermal conductivity.

### 3.3. Mechanical Properties and TEM Analysis of as-Aged Alloys

Figure 6 shows the stress–strain curves of the Al–Zn–Ca–Sc–Zr alloys after 3 h aging at 300 °C. The mechanical properties of the alloys are presented in Table 3. Although the AlZn3Ca3Sc0.3 alloy possesses a larger average grain size (Figure 3a), it demonstrated superior mechanical properties compared to alloys comprising Zr. The substitution of Sc with Zr led to a decrease in tensile yield strength (TYS) from 217.5 to 137.2 MPa and ultimate tensile strength (UTS) from 269.0 to 206.3 MPa, while elongation at fracture (El) increased from 4.6 to 7.1%. It is evident that the alteration in grain size does not serve as a pivotal factor influencing strengthening. The observed strength reduction with increasing Zr content correlates with decreased Sc concentration in the alloys. These findings are consistent with those reported by Yang J. et al. [55], who found that Zr additions exceeding 0.1 wt% in Al–6.5Mg–0.50Mn–0.25Sc–Zr alloys lead to the formation of coarse primary Al_3_(Sc,Zr) particles at higher Zr concentrations, which diminishes the effectiveness of precipitation strengthening. As illustrated in Figure 1e, the proportion of primary D0_23_-Al_3_(Sc,Zr) crystals in the studied alloys increases, while the proportion of phase L_12_-Al_3_(Sc,Zr) decreases. This, in turn, could also lead to a decrease in the efficiency of precipitation strengthening. Nevertheless, all investigated alloys exhibit significantly enhanced strength compared to the base Al–3 wt% Zn–3 wt% Ca alloy. For example, the TYS of all alloys increased by at least two times compared to the initial values [24]. The TYS of patented Tesla alloys was 90–160 MPa; the alloys studied in this work are not inferior to them and even surpass them with 0.2 wt% Sc [53].

Figure 7 shows TEM images of the AlZn3Ca3Sc0.3 and AlZn3Ca3Sc0.2Zr0.1 alloys that were aged at 300 °C for 3 h. The images in Figure 7a–c were obtained along the ${\left[01\bar{1}\right]}_{Al}$ zone axis, while those in Figure 7d–f were obtained along the ${\left[001\right]}_{Al}$ zone axis. The diffraction patterns revealed that the precipitated phases correspond to an L1_2_-ordered structure. The L1_2_-Al_3_Sc and L1_2_-Al_3_(Sc,Zr) precipitates have a nearly spherical shape, and the precipitates exhibit a “coffee bean” contrast (Figure 7a), indicating their coherence with the Al matrix. The obtained images provide evidence that the high hardness and strength of the alloys are attributable to the strengthening resulting from the precipitation of phase with L1_2_ structure.

Figure 8 shows the effect of aging time and temperature on the precipitate diameter according to the calculation results in Thermo-Calc for the Al-Zr-Sc alloys (AlSc0.3 and AlSc0.2Zr0.1). The Ca and Zn was excluded from the calculation due to the calculation errors connected with shortcoming of used software. The calculation indicates that the diameter of L1_2_-Al_3_Sc/L1_2_-Al_3_(Sc,Zr) precipitates should increase with increasing aging temperature and holding time. According to calculations performed, the L1_2_-Al_3_Sc/L1_2_-Al_3_(Sc,Zr) precipitates after aging at 300 °C for 3 h should have a diameter of approximately 14 nm for both AlSc0.3 and AlSc0.2Zr0.1 alloys. However, the experimentally determined average precipitate diameter was found to be considerably smaller. The average diameter determined by TEM images of L1_2_-Al_3_Sc/L1_2_-Al_3_(Sc,Zr) precipitates in the AlZn3Ca3Sc0.3 and AlZn3Ca3Sc0.2Zr0.1 alloys aged at 300 °C for 3 h was 8.56 ± 0.39 and 2.39 ± 0.04 nm, respectively. Furthermore, it is evident that the substitution of Sc with Zr resulted in a decrease in the precipitate diameter. Fuller et al. [56] investigated the evolution of the size of precipitates Al_3_(Sc_1−*x*_Zr*_x_*) as a function of aging time at 300 °C and found that Al_3_(Sc,Zr) precipitates have greater resistance to coarsening than Al_3_Sc.

### 3.4. Effect of Sc and Zr on Corrosion Properties of as-Aged Alloys

Figure 9 presents the corrosion rates of as-aged Al–Zn–Ca–Sc–Zr alloys determined by immersion corrosion testing. The corrosion rate of the alloy with 0.3 wt% Sc was approximately 0.022 mm/year. Partial substitution of Sc with Zr reduced corrosion rates to 0.002–0.006 mm/year, bringing it closer to that of the base Al–3 wt% Zn–3 wt% Ca alloy (0.005 mm/year) [24]. According to the graph, variations in Zr content within 0.05–0.2 wt% did not substantially affect corrosion rates, indicating that even minimal Zr additions (0.05 wt%) effectively enhance corrosion resistance.

Cross-sections of the samples after immersion corrosion testing revealed that the majority of the surfaces exhibited minimal alterations, with instances of corrosion cavities being infrequent. The cross-sectional images of the as-aged alloys specimens with typical corrosion cavities and the average corrosion cavity depth (ACC) after immersion corrosion testing are presented in Figure 10. It is evident that local corrosion damage developed from the outer to the inner regions of the alloy along the eutectic area in alloy structure. As previously established in [57], Al–Ca–Si alloys exhibit enhanced corrosion resistance, attributable to their refined eutectic structures, which impede the penetration of corrosion cavity depth into the alloy. In the present study, the eutectic lamellas of the (Al,Zn)_4_Ca phase, acting as an anode, likely played a pivotal role in the high corrosion resistance of the alloys. According to the findings reported in [58,59,60], Al_3_Sc and Al_3_(Sc,Zr) particles function as cathodes in relation to the Al matrix, thereby facilitating the progression of pitting corrosion. However, Cavanaugh et al. [58] found that the kinetics of the cathodic reaction of Al_3_Sc/Al is sluggish and should not pose a major corrosion risk. The AlZn3Ca3Sc0.3 alloy exhibited the average corrosion depth cavity of 27.8 ± 8.0 μm. In comparison, the alloy with 0.2 wt% Zr demonstrated average corrosion cavity depth of 10.1 ± 1.1 μm. The observed discrepancy in the corrosion cavity depth indicates that increasing the ratio of Zr can enhance the corrosion resistance of the Al–Zn–Ca–Sc–Zr alloys to local corrosion. Furthermore, when alloying with 0.05 wt% Zr (Figure 10b), although substantial corrosion cavities remain, their prevalence on the specimen’s surface diminishes, as substantiated by the diminished corrosion rate of the alloy (Figure 9).

Typical polarization curves obtained for as-aged Al–Zn–Ca–Sc–Zr alloys in a solution of 5.7% NaCl + 0.3% H_2_O_2_ with distilled water are shown in Figure 11. The corrosion characteristics of the alloys, including open circuit potential (OCP), the corrosion potential (E_corr_), and pitting potential (E_pit_), were obtained (Table 4). The OCP evolution was monitored over time for all specimens prior to electrochemical testing, with results presented in Figure 11a. The OCP values were found to stabilize at approximately −950 mV for all alloys, indicating that the corrosion resistance characteristics of the investigated compositions were approximately similar. Figure 11b shows the potentiodynamic polarization curves. For all alloys, the cathodic branch of the polarization curve manifests conventional characteristics of diffusion control within a non-mixing electrolyte. The anode branch of the polarization curve for all alloys displays a passive region (∆E_passive_), accompanied by a subsequent transition to pitting at the E_pit_. Additionally, it has been observed that alloys with a higher amount of Zr exhibit more positive E_pit_, indicating that increasing the ratio of Zr contributes to enhancing the stability of the passive film on the surface of the specimens. The E_corr_ of the alloys ranges from −1.17 to −1.21 V, and the addition of Zr can shift the E_corr_ slightly to the more negative values. A faster transition to the anodic region and broader passive region of Zr containing alloys in comparison to alloy without Zr (AlZn3Ca3Sc0.3) indicates faster passivation, which leads to better corrosion resistance in this electrolyte. The distinction between OCP and E_corr_ is attributable to the passive film that is formed in this electrolyte.

Thus, the corrosion tests demonstrated that addition of Zr has positive effect on the corrosion resistance of Al–Zn–Ca–Sc–Zr alloys. Kozlova et al. [61] also discovered that Al_3_(Sc,Zr) particles with a substantial Zr proportion exert a less pronounced effect on the corrosion rate in comparison to Al_3_Sc. The authors attribute this phenomenon to a decrease in the number of Al_3_(Sc,Zr) particles caused by a decrease in the total content of Sc and Zr (at%) when replacing Sc with Zr due to the difference in atomic mass between these two elements. At the same time in this work, it was established that the L1_2_-Al_3_Sc/L1_2_-Al_3_(Sc,Zr) precipitates in AlZn3Ca3Sc0.2Zr0.1 alloy is more than three times finer than in AlZn3Ca3Sc0.3 alloy (Figure 7 and Figure 8) and that also can be a reason for the higher corrosion resistance of alloys with Zr addition.

### 3.5. Thermal Conductivity of Al–3wt%Zn–3wt%Ca–0.3wt%Sc Alloy

The alloy exhibiting the highest thermal conductivity in accordance with Equation (1) was selected for the additional investigation of influence of temperature on thermal conductivity. Equation (2) was used to determine λ(T). The thermal properties and density of the AlZn3Ca3Sc0.3 alloy after aging are shown in Figure 12. As the temperature rises, thermal diffusivity undergoes a progressive decrease, and conversely, heat capacity experiences an increase. Concurrently, a slight decrease in the density is exhibited, attributable to thermal expansion. The interplay between these parameters leads to a negligible variation in thermal conductivity calculated in accordance with Equation (2) across the studied temperature range. As the temperature increases from room temperature to 132 °C, the thermal conductivity of the alloys exhibits a slight increase, from 159.6 to 161.4 W/mK. A further increase in temperature to 300 °C results in a slight decrease in thermal conductivity, which reaches 159.0 W/mK. These changes remain within the 1% range, suggesting that thermal conductivity remains essentially constant across this temperature range. Furthermore, the calculated thermal conductivity is 14% lower than that obtained using Equation (1), which was 184.7 W/mK. The thermal diffusivity method using Equation (2) relies on multiple measurements (density, diffusivity, and heat capacity), each contributing to the cumulative error. In contrast, the Smith–Palmer Equation (1) was specifically derived for aluminum alloys; its coefficients were calibrated using extensive data on industrial alloys. Therefore, results obtained via the Smith–Palmer method are likely more accurate.

## 4. Conclusions

In this study, the effect of Sc and Zr (total content: 0.3 wt%) on the Al–3 wt% Zn–3 wt% Ca alloy properties was investigated. The following conclusions can be drawn from the present investigation:The microstructure of the as-cast alloys consists of α-Al dendrites and refined lamellas of α-Al + (Al,Zn)_4_Ca eutectic. Sc was found to be uniformly distributed throughout the aluminum matrix, while Zr was concentrated in the center of dendritic cells. All alloys demonstrated coarse grain structure, but the addition of Zr lead to a little decrease in the grain size. The minimal grain size of 338 μm was observed for the alloy with 0.2 wt% Sc and 0.05 wt% Zr.In the as-cast state, the substitution of scandium with zirconium did not result in a substantial alteration of the alloys’ hardness and was approximately 50 HB. However, this substitution did lead to a decline in the thermal conductivity of the alloys, with a decrease from 167.6 to 158.9 W/mK (calculated via Smith–Palmer equation).The most suitable heat treatment regime was determined to be the aging treatment at 300 °C for 3 h. This regime resulted in a substantial enhancement of the alloys hardness to 93 HB and thermal conductivity obtained, via the Smith-Palmer equation, to ~185 W/mK. However, an increase in the Zr content led to a decrease in hardness and thermal conductivity in as-aged alloys.The Al–Zn–Ca–Sc–Zr alloys exhibit high mechanical properties in as-aged state due to the precipitation of strengthening Al_3_Sc/Al_3_(Sc,Zr) phase during the aging process. However, substituting Sc with Zr results in a decrease in the UTS of the alloys from 269 to 206 MPa and an increase in their El from 4.6 to 7.1%. The highest strength was obtained in the alloy with 0.3% Sc.The immersion corrosion test results demonstrated that replacing Sc with Zr can reduce the corrosion rate of as-aged alloys from 0.022 to 0.002–0.006 mm/year and reduce the average corrosion cavity depth from 27.8 ± 8.0 μm to 10.1 ± 1.1 μm. Electrochemical corrosion testing has demonstrated that the partial substitution of Sc with Zr in alloy compositions enhances the resistance of the alloys to pitting corrosion.The thermal conductivity of the AlZn3Ca3Sc0.3 alloy in as-aged condition, which was determined using the thermal diffusivity equation, shows exceptional stability (±1%) across 25–300 °C.

## Figures and Tables

**Figure 1 materials-18-05680-f001:**
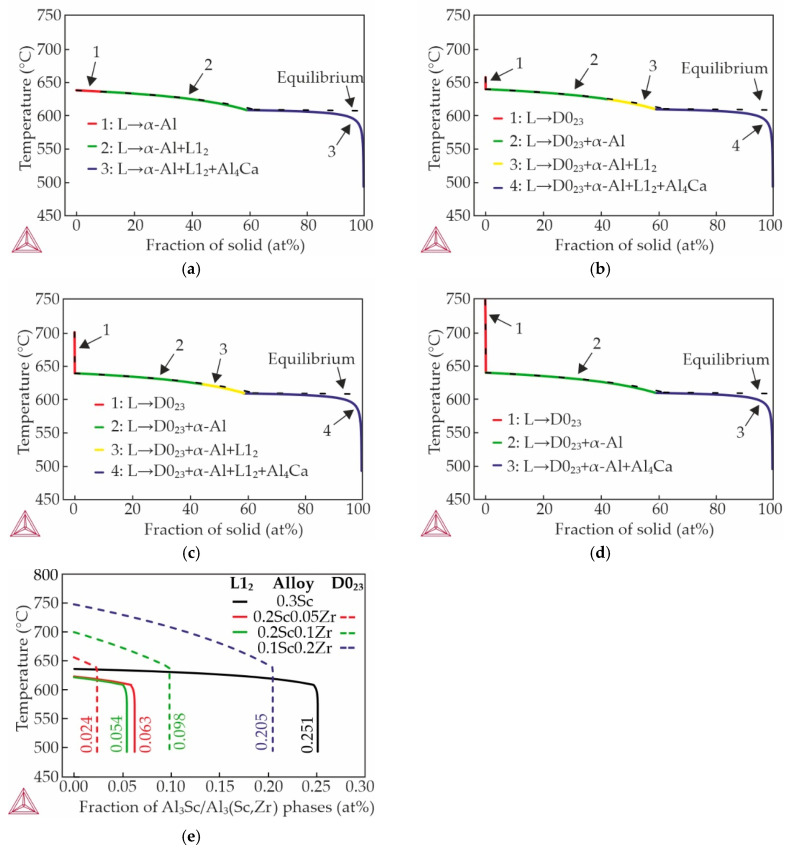
Solidification paths of alloys: (**a**) AlZn3Ca3Sc0.3, (**b**) AlZn3Ca3Sc0.2Zr0.05, (**c**) AlZn3Ca3Sc0.2Zr0.1, (**d**) AlZn3Ca3Sc0.1Zr0.2, and (**e**) fraction of L1_2_–Al_3_Sc/Al_3_(Sc,Zr) and D0_23_–Al_3_(Sc,Zr) phases.

**Figure 2 materials-18-05680-f002:**
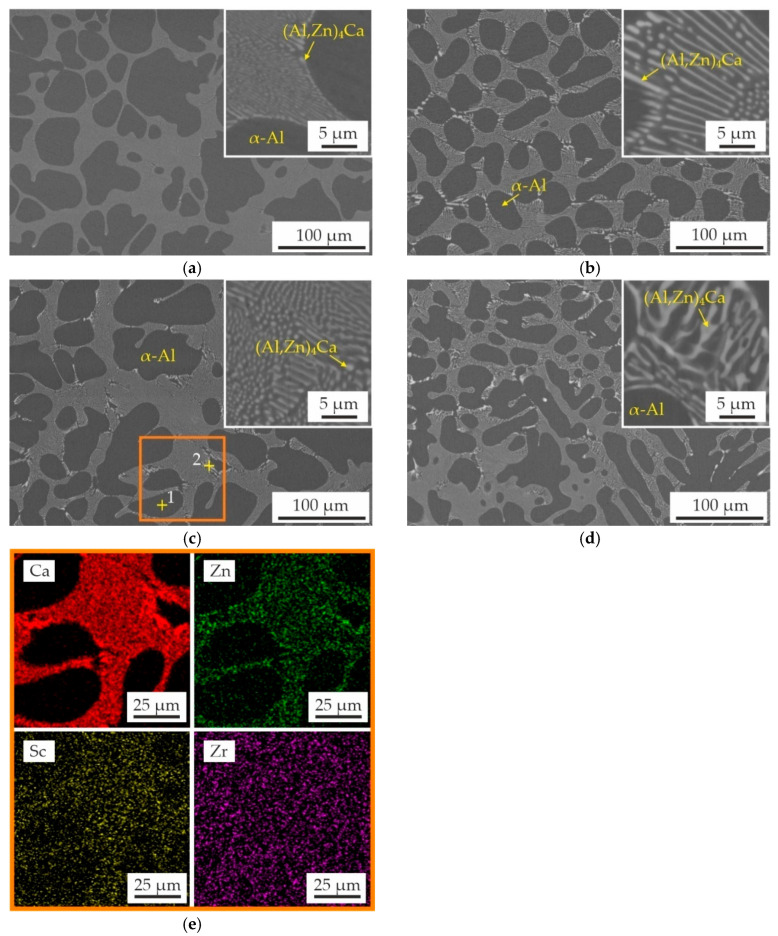
As-cast microstructure (SEM) of (**a**) AlZn3Ca3Sc0.3, (**b**) AlZn3Ca3Sc0.2Zr0.05, (**c**) AlZn3Ca3Sc0.2Zr0.1, (**d**) AlZn3Ca3Sc0.1Zr0.2 alloys, and (**e**) EDS maps showing alloying elements distribution in selected area of AlZn3Ca3Sc0.2Zr0.1 alloy (orange rectangle in (**c**)).

**Figure 3 materials-18-05680-f003:**
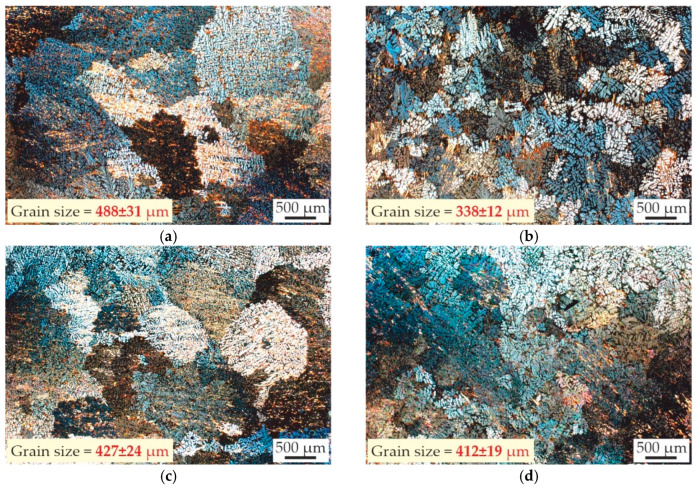
Macrostructure of as-cast alloys: (**a**) AlZn3Ca3Sc0.3, (**b**) AlZn3Ca3Sc0.2Zr0.05, (**c**) AlZn3Ca3Sc0.2Zr0.1, (**d**) AlZn3Ca3Sc0.1Zr0.2.

**Figure 4 materials-18-05680-f004:**
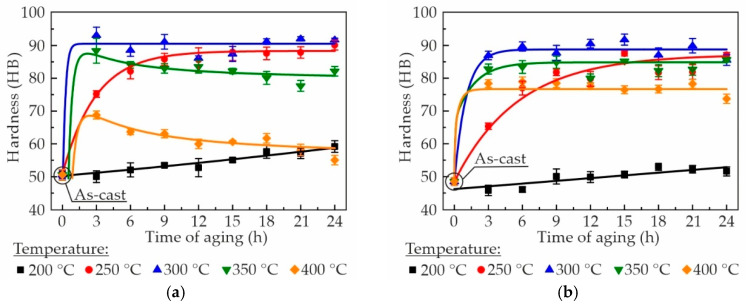
Hardness evolution of the Al–Zn–Ca–Sc–Zr alloys: (**a**) AlZn3Ca3Sc0.3, (**b**) AlZn3Ca3Sc0.2Zr0.05, (**c**) AlZn3Ca3Sc0.2Zr0.1, (**d**) AlZn3Ca3Sc0.1Zr0.2, during aging treatment at 200–400 °C.

**Figure 5 materials-18-05680-f005:**
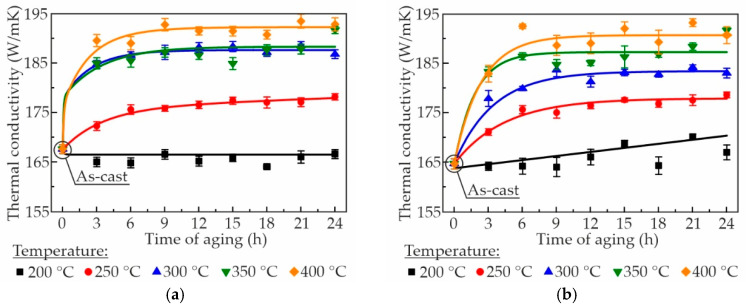
Thermal conductivity evolution of the Al–Zn–Ca–Sc–Zr alloys: (**a**) AlZn3Ca3Sc0.3, (**b**) AlZn3Ca3Sc0.2Zr0.05, (**c**) AlZn3Ca3Sc0.2Zr0.1, (**d**) AlZn3Ca3Sc0.1Zr0.2, during aging treatment at 200–400 °C.

**Figure 6 materials-18-05680-f006:**
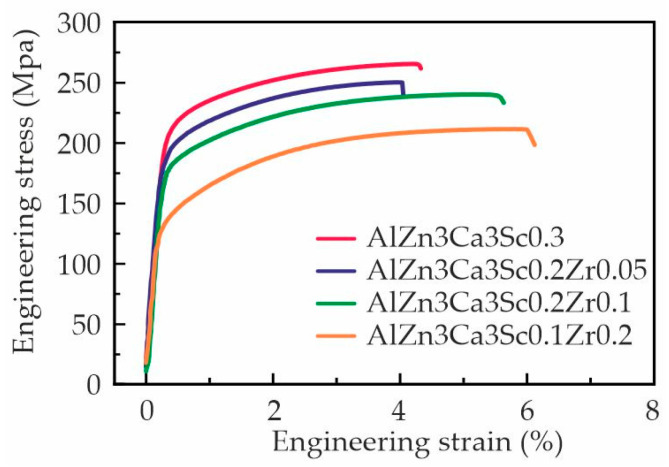
The engineering stress–strain curves of the aged during 3 h at 300 °C Al–Zn–Ca–Sc–Zr alloys.

**Figure 7 materials-18-05680-f007:**
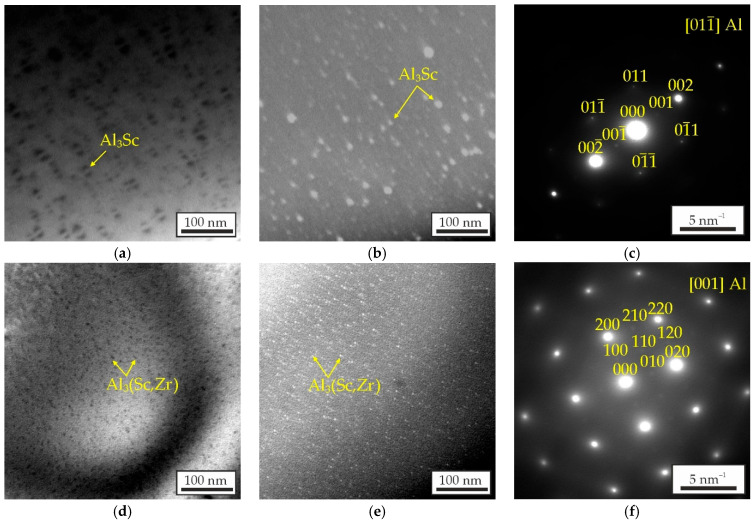
TEM (**a**,**d**) bright field images, (**b**,**e**) dark field image, and (**c**,**f**) selected area diffraction patterns of aged for 3 h at 300 °C (**a**–**c**) AlZn3Ca3Sc0.3 and (**d**–**f**) AlZn3Ca3Sc0.2Zr0.1 alloys.

**Figure 8 materials-18-05680-f008:**
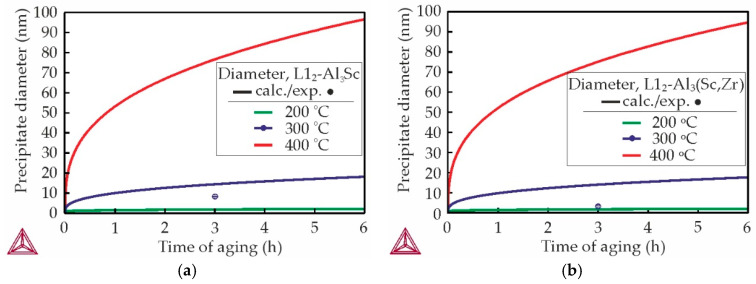
Calculated diameter of Al_3_Sc/Al_3_(Sc,Zr) precipitates in Al-Sc-Zr alloys after aging at 200–400 °C (solid lines): (**a**) Al–0.3wt%Sc and (**b**) Al–0.2wt%Sc–0.1wt%Zr. Experimental average diameter of precipitates in (**a**) AlZn3Ca3Sc0.3 and (**b**) AlZn3Ca3Sc0.2Zr0.1 alloy following 3 h aging at 300 °C, as determined by TEM (dots).

**Figure 9 materials-18-05680-f009:**
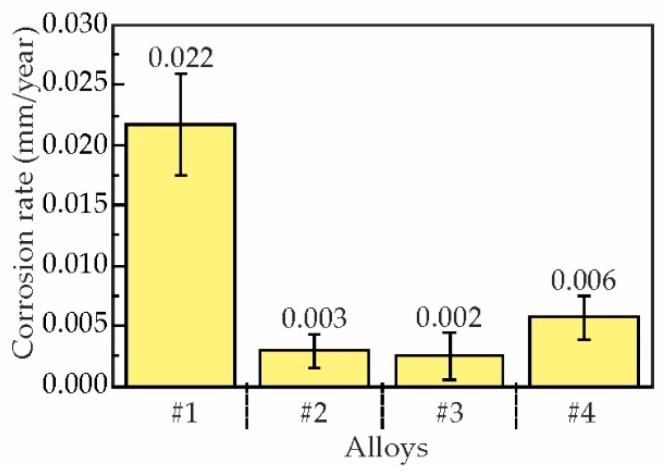
Corrosion rate of the as-aged Al–Zn–Ca–Sc–Zr alloys (300 °C, 3 h) determined after 15-day immersion test in 5.7% NaCl + 0.3% H_2_O_2_ solution: #1—AlZn3Ca3Sc0.3, #2—AlZn3Ca3Sc0.2Zr0.05, #3—AlZn3Ca3Sc0.2Zr0.1, #4—AlZn3Ca3Sc0.1Zr0.2.

**Figure 10 materials-18-05680-f010:**
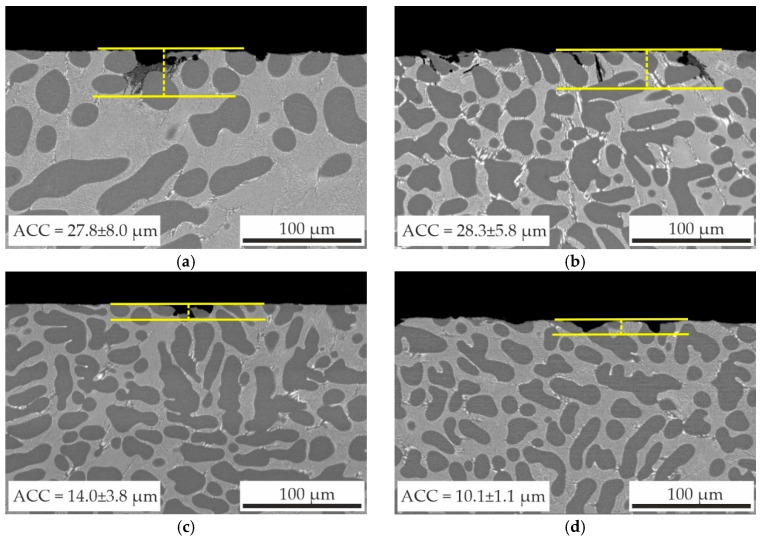
Cross-sectional microstructure and average corrosion cavity depths (ACD) of as-aged specimens of alloys: (**a**) AlZn3Ca3Sc0.3, (**b**) AlZn3Ca3Sc0.2Zr0.05, (**c**) AlZn3Ca3Sc0.2Zr0.1, and (**d**) AlZn3Ca3Sc0.1Zr0.2 after 15-day immersion test in 5.7% NaCl + 0.3% H_2_O_2_ water solution. The yellow horizontal lines are markers indicating the typical depth of corrosion cavities.

**Figure 11 materials-18-05680-f011:**
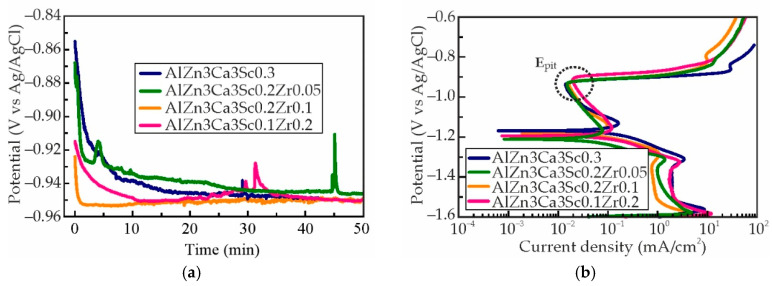
(**a**) Evolution of the OCP versus exposure time and (**b**) Polarization curves of as-aged Al–Zn–Ca–Sc–Zr alloys in 5.7% NaCl + 0.3% H_2_O_2_ solutions.

**Figure 12 materials-18-05680-f012:**
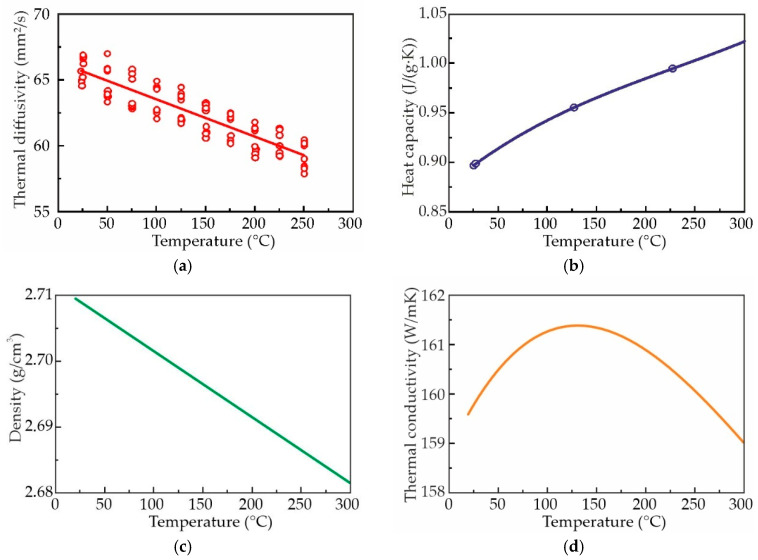
(**a**) Thermal diffusivity, (**b**) heat capacity, (**c**) density, and (**d**) thermal conductivity of the as-aged AlZn3Ca3Sc0.3 alloy.

**Table 1 materials-18-05680-t001:** Chemical compositions of investigated alloys.

Alloy	Element Content (wt%)				
	Al	Zn	Ca	Sc	Zr
AlZn3Ca3Sc0.3	Bal.	3.09	2.87	0.32	–
AlZn3Ca3Sc0.2Zr0.05	Bal.	3.13	2.99	0.20	0.06
AlZn3Ca3Sc0.2Zr0.1	Bal.	3.09	2.82	0.17	0.11
AlZn3Ca3Sc0.1Zr0.2	Bal.	3.29	2.93	0.12	0.23

**Table 2 materials-18-05680-t002:** Composition of structural constituents obtained by EDS in points presented in Figure 2c for AlZn3Ca3Sc0.2Zr0.1.

Points	Element Content (wt%)					Phase/Structure Constituents
	Al	Zn	Ca	Sc	Zr	
1	Bal.	1.11 ± 0.01	0.01 ± 0.01	0.16 ± 0.03	0.30 ± 0.08	α-Al
2	Bal.	10.62 ± 2.40	13.44 ± 3.88	0.13 ± 0.02	0.01 ± 0.01	α-Al + (Al,Zn)_4_Ca

**Table 3 materials-18-05680-t003:** Mechanical properties of the Al–Zn–Ca–Sc–Zr alloys with different Sc/Zr ratios after aging at 300 °C for 3 h.

Specimen Designation	TYS (MPa)	UTS (MPa)	El (%)
AlZn3Ca3Sc0.3	217.5 ± 5.8	269.0 ± 6.9	4.6 ± 1.0
AlZn3Ca3Sc0.2Zr0.05	195.2 ± 1.5	250.7 ± 2.0	4.5 ± 1.0
AlZn3Ca3Sc0.2Zr0.1	180.9 ± 4.8	235.3 ± 6.6	5.1 ± 2.2
AlZn3Ca3Sc0.1Zr0.2	137.2 ± 5.4	206.3 ± 6.1	7.1 ± 2.0

**Table 4 materials-18-05680-t004:** Electrochemical corrosion test data in 5.7% NaCl + 0.3% H_2_O_2_ water solution obtained for Al–Zn–Ca–Sc–Zr alloys with different Sc/Zr ratios after aging at 300 °C for 3 h.

Specimen Designation	OCP (V)	E_corr_ (V)	E_pit_ (V)	∆E_passive_ (V)
AlZn3Ca3Sc0.3	–0.950	–1.17	–0.934	0.190
AlZn3Ca3Sc0.2Zr0.05	–0.946	–1.21	–0.930	0.236
AlZn3Ca3Sc0.2Zr0.1	–0.950	–1.18	–0.925	0.209
AlZn3Ca3Sc0.1Zr0.2	–0.950	–1.19	–0.907	0.253

## Data Availability

The original contributions presented in this study are included in the article. Further inquiries can be directed to the corresponding author.

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
