# Peer review of "The Effect of Sc and Zr Additions on the Structure, Mechanical, and Corrosion Properties of a High Thermal Conductive Al–3%Zn–3%Ca Alloy"

_materials, 2025, doi:10.3390/ma18245680_

Round 1

Reviewer 1 Report

Comments and Suggestions for Authors

This manuscript  systematically investigates the effects of Sc and Zr additions on the microstructure, mechanical properties, thermal conductivity, and corrosion resistance of an Al-3Zn-3Ca alloy. The chosen topic possesses clear engineering relevance and scientific merit. The experimental design is sound, the data analysis is thorough, and the conclusions are clearly presented. The overall quality of the manuscript is high, and it is recommended for publication after **minor revisions**, pending modifications and improvements in the following aspects:

1.In the Introduction section, the core innovative aspects of this work should be more clearly articulated. This paper does not fully describe the latest research progress and reference the latest theories in this field, for example:(1), Materials Today Communications 48 (2025) 113606. https://doi.org/10.1016/j.mtcomm.2025.113606. (2),Journal of Materials Science  (2025) 1-23,https://doi.org/10.1007/s10853-025-11369-5. (3), Adv. Eng. Mater. 27 (2025) 2402052,  https://doi.org/10.1002/adem.202402052,and (4) CHIN J MECH ENG-EN.Volume38,Issue1,(2025) 38:81,https://doi.org/10.1186/s10033-025-01243-5. It is suggested to refer to the above research to help improve the quality of the article.

  1. A significant discrepancy (approximately 14%) exists between the thermal conductivity calculated using the Smith-Palmer equation (~185 W/mK) and the value measured via the laser flash analysis (LFA) method (~160 W/mK) in the text. Suggestion: The authors should discuss the potential reasons for this discrepancy (e.g., applicability of the formula, sample condition, measurement errors) and clearly state in the conclusion which method is more suitable for this type of alloy.
  2. Specify the strain rate parameter used in the tensile tests (e.g., as recommended by ASTM standards). This parameter significantly influences both strength and ductility data, and its inclusion would enhance the comparability of the results.
  3. The specific procedure for replenishing Hâ‚‚Oâ‚‚ during corrosion testing (e.g., the calculation method for replenishment quantity, means of concentration monitoring) should be clearly specified to ensure the reproducibility of corrosive environment stability.

Author Response

Comment 1: In the Introduction section, the core innovative aspects of this work should be more clearly articulated. This paper does not fully describe the latest research progress and reference the latest theories in this field, for example:(1), Materials Today Communications 48 (2025) 113606. https://doi.org/10.1016/j.mtcomm.2025.113606. (2),Journal of Materials Science  (2025) 1-23,https://doi.org/10.1007/s10853-025-11369-5. (3), Adv. Eng. Mater. 27 (2025) 2402052,  https://doi.org/10.1002/adem.202402052,and (4) CHIN J MECH ENG-EN.Volume38,Issue1,(2025) 38:81,https://doi.org/10.1186/s10033-025-01243-5. It is suggested to refer to the above research to help improve the quality of the article.

Response 1: The Introduction section has undergone modifications. In order to enhance the clarity and quality of the article, a review of recent research trends in high-thermal-conductivity alloys has been undertaken. A reference to the source has also been incorporated: (1), Materials Today Communications 48 (2025) 113606. https://doi.org/10.1016/j.mtcomm.2025.113606.

«Commercial casting aluminum alloys typically exhibit either high castability and strength or good thermal conductivity. This is attributed to the fact that aluminum alloys with good castability contain large amounts of alloying elements such as Si, Mg, and Cu. When present in high concentrations, these elements significantly impair the alloys' thermal conductivity [5,6]. A large number of studies have been dedicated to addressing this issue [7–11]. It has been demonstrated that heat treatment of alloys can improve their thermal conductivity [12–14]. However, this outcome is frequently inadequate in ensuring effective heat transfer, owing to the initially low thermal conductivity of commercial cast alloys [15]. Alternative approaches involve incorporating reinforcements with high intrinsic thermal conductivity, such as graphene nanoplatelets (~5000 W/mK), into aluminum matrix [16–18]. This method can significantly improve the electrical and thermal conductivity of aluminum alloys. For example, Andilab et al. [17] succeeded in increasing the electrical conductivity of cast A319 Al alloy from 27 to 30.2% IACS after adding 0.2 wt% graphene. However, this technique is not yet fully explored, and potential adverse effects on castability, essential for manufacturing thin-walled die-cast components, remain a concern. Therefore, a key strategy for developing aluminum alloys with high thermal conductivity involves the selection of alloying elements with extremely low solubility in the α-Al solid solution or control the overall solubility of alloying elements in the alloy [19]».

In addition, the novelty of this work was explained in more detail.

«A substantial body of research has been dedicated to investigating the impact of Sc and Zr on the structure and mechanical properties of aluminum alloys. However, a comprehensive analysis of the influence of these elements on Al–Zn–Ca alloys, as well as their effect on the alloys' thermal conductivity, remains absent from the extant published literature, indicating a significant gap and novelty».

Comment 2: A significant discrepancy (approximately 14%) exists between the thermal conductivity calculated using the Smith-Palmer equation (~185 W/mK) and the value measured via the laser flash analysis (LFA) method (~160 W/mK) in the text. Suggestion: The authors should discuss the potential reasons for this discrepancy (e.g., applicability of the formula, sample condition, measurement errors) and clearly state in the conclusion which method is more suitable for this type of alloy.

Response 2: In general, both methods are indirect. The method that uses the thermal diffusivity equation is based on a larger number of measurements (density, thermal diffusivity, heat capacity), and each measurement introduces error. The Smith and Palmer method has the advantage that the resulting equation was developed specifically for aluminum alloys. A large body of data on industrial aluminum alloys was used to calculate the equation's coefficients. For this reason, values obtained using the second (Smith and Palmer equation) method are likely more accurate. At the same time, the first method (the thermal diffusivity equation) allows thermal conductivity to be determined at temperatures above room temperature since the alloy will operate at elevated temperatures. The text has been amended to place greater emphasis on the role of the methods used.

The changes in the 3.5 section are as follows:

«The alloy exhibiting the highest thermal conductivity in accordance with Equation (1) was selected for the additional investigation of influence of temperature on thermal conductivity. Equation (2) was used to determine λ(T)…

The thermal diffusivity method (2) relies on multiple measurements (density, diffusivity, heat capacity), each contributing to the cumulative error. In contrast, the Smith–Palmer equation (1) was specifically derived for aluminum alloys; its coefficients were calibrated using extensive data on industrial alloys. Therefore, results obtained via the Smith–Palmer method are likely more accurate».

Comment 3: Specify the strain rate parameter used in the tensile tests (e.g., as recommended by ASTM standards). This parameter significantly influences both strength and ductility data, and its inclusion would enhance the comparability of the results.

Response 3: The tests were conducted at a constant strain rate of 0.002 s–1. This information was added to Materials and Methods section of the work.

Comment 4: The specific procedure for replenishing Hâ‚‚Oâ‚‚ during corrosion testing (e.g., the calculation method for replenishment quantity, means of concentration monitoring) should be clearly specified to ensure the reproducibility of corrosive environment stability.

Response 4: In the Materials and Methods section, quantitative data regarding the H2O2 utilized has been incorporated (0.3% H2O2 of the testing solution volume). This step in the test methodology has been incorporated in accordance with the Russian standard for a comparable test – GOST 9.913-90. Unified system of corrosion and aging protection. Aluminum, magnesium, and their alloys. Methods of accelerated corrosion tests. The reference was not provided because it is in Russian language.

Reviewer 2 Report

Comments and Suggestions for Authors

1) Introduction is very brief with a limited number of references. You need to expand the introduction, include more relevant works on the microalloying of Al-Zn-Ca and similar Al alloys.

2) Please explain the novelty of this work in detail.

3) You need to add the hardness, mechanical properties, thermal conductivity and corrosion properties of the Al-Zn-Ca alloy, it is the reference and it is important to understand if the various alloying combinations offer a significant improvement.

4) Please add a table with the main electrochemical values from the polarisation curves like Ecorr, Epit, icorr. This will help expand the discussion on the corrosion performance of the different alloys.

5) What is the porosity of the different studied alloys?

6) You need to compare the results with other popular Al alloys used in similar applications.  

Author Response

Comment 1: Introduction is very brief with a limited number of references. You need to expand the introduction, include more relevant works on the microalloying of Al-Zn-Ca and similar Al alloys.

Response 1: The Introduction section has been expanded. In order to enhance quality of the article, a review of recent research trends in high-thermal-conductivity alloys has been undertaken.

«Commercial casting aluminum alloys typically exhibit either high castability and strength or good thermal conductivity. This is attributed to the fact that aluminum alloys with good castability contain large amounts of alloying elements such as Si, Mg, and Cu. When present in high concentrations, these elements significantly impair the alloys' thermal conductivity [5,6]. A large number of studies have been dedicated to addressing this issue [7–11]. It has been demonstrated that heat treatment of alloys can improve their thermal conductivity [12–14]. However, this outcome is frequently inadequate in ensuring effective heat transfer, owing to the initially low thermal conductivity of commercial cast alloys [15]. Alternative approaches involve incorporating reinforcements with high intrinsic thermal conductivity, such as graphene nanoplatelets (~5000 W/mK), into aluminum matrix [16–18]. This method can significantly improve the electrical and thermal conductivity of aluminum alloys. For example, Andilab et al. [17] succeeded in increasing the electrical conductivity of cast A319 Al alloy from 27 to 30.2% IACS after adding 0.2 wt% graphene. However, this technique is not yet fully explored, and potential adverse effects on castability, essential for manufacturing thin-walled die-cast components, remain a concern. Therefore, a key strategy for developing aluminum alloys with high thermal conductivity involves the selection of alloying elements with extremely low solubility in the α-Al solid solution or control the overall solubility of alloying elements in the alloy [19]».

Comment 2: Please explain the novelty of this work in detail.

Response 2: The novelty of this work was explained in more detail. The Introduction of the work has been adjusted.

«A substantial body of research has been dedicated to investigating the impact of Sc and Zr on the structure and mechanical properties of aluminum alloys. However, a comprehensive analysis of the influence of these elements on Al–Zn–Ca alloys, as well as their effect on the alloys' thermal conductivity, remains absent from the extant published literature, indicating a significant gap and novelty».

Comment 3: You need to add the hardness, mechanical properties, thermal conductivity and corrosion properties of the Al-Zn-Ca alloy, it is the reference and it is important to understand if the various alloying combinations offer a significant improvement.

Response 3: The paper was updated with the necessary data on the Al–3 wt% Zn–3 wt% Ca alloy. A reference to an earlier paper presenting more research on the properties of Al–Zn–Ca ternary alloys was also provided.

Thermal conductivity and hardness:

« This heat treatment achieved dual objectives: it restored thermal conductivity to levels approaching the base Al–3 wt% Zn–3 wt% Ca alloy (194 W/mK [24]), while increasing hardness by at least 31 HB relative to the base alloy (41 HB [24])».

Strength:

« Nevertheless, all investigated alloys exhibit significantly enhanced strength compared to the base Al–3 wt% Zn–3 wt% Ca alloy. For example, the TYS of all alloys increased by at least 2 times compared to the initial values [24]».

Corrosion:

«Partial substitution of Sc with Zr reduced corrosion rates to 0.002–0.006 mm/year, bringing it closer to that of the base Al–3 wt.% Zn–3 wt.% Ca alloy (0.005 mm/year [24])».

Comment 4: Please add a table with the main electrochemical values from the polarisation curves like Ecorr, Epit, icorr. This will help expand the discussion on the corrosion performance of the different alloys.

Response 4: Section 3.4 has been updated with a Table 4 containing the basic data, excluding icorr (corrosion current). You can't easily calculate icorr with simple Tafel extrapolation when diffusion controls because the reaction rate is limited by mass transport (species moving to the surface), not just electron transfer kinetics, causing the polarization curve to flatten out (limiting current), not showing a clear straight Tafel slope, making extrapolation inaccurate.

Comment 5: What is the porosity of the different studied alloys?

Response 5: The structural analysis of the studied samples revealed no evidence of porosity. The solidification behavior of the alloys and their casting properties were the focus of another unpublished study. The study concluded that the alloys under investigation are not prone to porosity formation due to their narrow freezing range.

Comment 6: You need to compare the results with other popular Al alloys used in similar applications. 

Response 6: The article compared the alloys that were studied with industrial die-casting alloys with high thermal conductivity from Tesla. For example: «The TYS of patented Tesla alloys was 90-160 MPa; the alloys studied in this work are not inferior to them and even surpass them with 0.2 wt% Sc [54]» or « Furthermore, the thermal conductivity of the as-aged AlZn3Ca3Sc0.3 alloy (184.7 W/mK) is only slightly inferior that of the Al–3.6Si–0.4Mg–0.03Ti alloy (T5 – 225 °C, 2h) used by Tesla (~190.7 W/mK using Equation (1)) [54]».